# Unidimensional ACGAN Applied to Link Establishment Behaviors Recognition of a Short-Wave Radio Station

**DOI:** 10.3390/s20154270

**Published:** 2020-07-31

**Authors:** Zilong Wu, Hong Chen, Yingke Lei

**Affiliations:** College of Electronic Countermeasures, National University of Defense Technology, Hefei 230037, China; wuzilong@nudt.edu.cn (Z.W.); ch2sun@mail.ustc.edu.cn (H.C.)

**Keywords:** unidimensional ACGAN, signal recognition, data augmentation, link establishment behaviors, DenseNet, short-wave radio station

## Abstract

It is difficult to obtain many labeled Link Establishment (LE) behavior signals sent by non-cooperative short-wave radio stations. We propose a novel unidimensional Auxiliary Classifier Generative Adversarial Network (ACGAN) to get more signals and then use unidimensional DenseNet to recognize LE behaviors. Firstly, a few real samples were randomly selected from many real signals as the training set of unidimensional ACGAN. Then, the new training set was formed by combining real samples with fake samples generated by the trained ACGAN. In addition, the unidimensional convolutional auto-coder was proposed to describe the reliability of these generated samples. Finally, different LE behaviors could be recognized without the communication protocol standard by using the new training set to train unidimensional DenseNet. Experimental results revealed that unidimensional ACGAN effectively augmented the training set, thus improving the performance of recognition algorithm. When the number of original training samples was 400, 700, 1000, or 1300, the recognition accuracy of unidimensional ACGAN+DenseNet was 1.92, 6.16, 4.63, and 3.06% higher, respectively, than that of unidimensional DenseNet.

## 1. Introduction

In the field of electronic reconnaissance, only a few Link Establishment (LE) behaviors signals of short-wave radio stations can be detected by non-collaborative sensors. Therefore, unidimensional ACGAN is utilized to get more LE behavior signals, avoiding the problem of lacking a large number of samples to train neural network. Actually, researchers in the military field are very concerned about how to recognize LE behaviors of non-collaborative radio stations, which help the researchers infer network topology of these radio stations. For example, if we find that the behavior of a radio station is Call behavior, some other radio stations that communicate with the radio station appear after a period of time. Hence, all the above radio stations belong to the same communication network. We can also infer how many radio stations are in the current communication network by analyzing the newly emerged electromagnetic signals.

The short-wave radio station refers to wireless communication equipment, of which the working frequency is 3–30 MHz. The most classic way of communication in the military field is to use a short-wave radio station due to its simple equipment, low power, and mature technologies. At present, most of the commands among the brigade, the battalion, and the company are transmitted via a short-wave radio station. Therefore, research on the LE behaviors of short-wave radio stations is of great significance to intelligence reconnaissance. The LE behaviors of a short-wave radio station are a kind of communication behavior of a radio station, which means a radio station starts a specific communication for different purposes. In fact, the seven kinds of LE signals correspond to seven kinds of LE behaviors consisting of Call behavior, Handshake behavior, Notification behavior, Time Offset behavior, Group Time Broadcast behavior, Broadcast behavior, and Scanning Call behavior [1]. Meaningfully, all research in military area desires to acquire the topology of the network where the radio station is located, and the status of the radio station as a node in the communication network, which could be facilitated by LE behavior recognition. For example, if a radio station frequently conducts Call behavior, the tactical status of the radio station is very important, and the owner is likely to be the commander in their organization. On the other hand, if a radio station seldom conducts Call behavior, the radio station may have a low tactical status. However, only radio LE behavior signals can be collected by non-collaborative sensors, which means there is no help for protocol standard of the radio station. Moreover, only a small number of LE behavior signals can be acquired from enemy radio stations, which increases the difficulty of LE behavior recognition. Limited to the unknown communication protocol standard and only a few collected labeled signals, LE behaviors can still be recognized by using the proposed algorithm in this work to directly process physical layer signals.

At present, the research on LE behavior recognition of radio stations at home and abroad is in the initial stage. Research has been done [2,3,4,5] on the communication behaviors of radio signals, but they all differ from a short-wave radio station’s LE behavior recognition. The research done in [6] uses a novel improved unidimensional DenseNet to recognize the LE behaviors of a short-wave radio station, and the whole recognition process does not need the help of communication protocol standard, avoiding complex signal feature transformation. However, when there are only a few samples with labels, the improved unidimensional DenseNet recognition accuracy needs to be further improved. In the actual electronic countermeasure environment, only a small amount of LE behavior signals can be obtained. In view of this special case, further research is needed. In terms of the implementation issue, it is possible to generalize current research results to the short wave radio station with constraints by combining the technologies in intelligent control and ideas in this work [7,8].

The Generative Adversarial Network (GAN) [9] can generate fake samples that are very similar to the real samples and then achieve the purpose of data augmentation, which indicates that GAN has great potential to solve the problem of a few LE behavior signals. The role of GAN can be roughly divided into style transfer [10,11,12,13] and data augmentation [14,15,16] according to application scenarios. In the field of style transfer, the pix2pix [17] is of epoch-making significance. Isola et al. realized the style transfer of paired images through pix2pix rather than simple pixel-to-pixel mapping. BicycleGAN further improved the performance of pix2pix [18]. CycleGAN [19] and DiscoGAN [20] are able to realize style transfer without using pairs of images. In the field of data augmentation, the research of GAN mainly focuses on the improvement of network structure. On the basis of the GAN model, Conditional Generative Adversarial Network (CGAN) [21,22] adds additional conditional information to the generator and discriminator to guide the training of the network model, and finally CGAN can generate samples corresponding to the specified labels. Energy-based Generative Adversarial Network (EBGAN) [23] introduced the concept and method of energy into GAN and regarded the discriminator as an energy function. ACGAN [24,25] added an auxiliary classifier to the output of the discriminator to improve the performance of GAN, and ACGAN also proposed using the class of each sample to update and improve the loss function, which significantly improved the performance of the network model. In the field of LE behavior recognition of short-wave radio stations, ACGAN can generate some labeled signals according to a small number of labeled signals, which achieves the purpose of data augmentation. However, the original ACGAN model is only applicable to the field of computer vision. Therefore, in this paper, unidimensional ACGAN is proposed to achieve data augmentation of unidimensional LE behavior signals.

Aiming at solving the problem that there are only a few LE behavior signals with labels of short-wave radio stations, a new unidimensional ACGAN is proposed to acquire more LE behavior signals. The following is the overall idea of this work: According to the short-wave communication protocol standard (MIL-STD-188-141B), seven kinds of LE behavior signals are simulated, and then these signals are used to verify the feasibility and effectiveness of the proposed algorithm model. Firstly, a small number of real LE behavior signals were randomly selected to train unidimensional ACGAN. In order to explore the effectiveness of unidimensional ACGAN, a new unidimensional Convolutional Auto-Encoder (CAE) which was used to demonstrate the deep features distribution of these generated samples was proposed. Then, the generated samples were mixed with the initial training samples to form a new training set, and the new training set was used to train the recognition network model. Finally, the LE behaviors of a short-wave radio station were recognized based on a small number of labeled samples. The whole training and recognition process of algorithm model did not need the help of communication protocol, which met the demand of real electronic countermeasures. Meanwhile, it also showed that the proposed algorithm model had the value of practical application.

Our main contributions are as follows:A method based on ACGAN+DenseNet was proposed to recognize radio stations’ LE behaviors without the communication protocol standard, which means a lot in the filed in the military field;A new ACGAN called unidimensional ACGAN was presented to generate more LE behavior signals. The presented ACGAN was able to directly process and generate unidimensional electromagnetic signals, while the original ACGAN is mostly used in the field of computer vision rather than unidimensional signals;We used a unidimensional Convolutional Auto-Encoder to represent deep features of the generated samples, which provided a novel way to verify the reliability of ACGAN when applied in the generation of electromagnetic signals.

The idea adopted in this work provides a reference for research on communication behaviors of non-collaborative radio stations. Once we have mastered the communication behaviors of radio stations belonging to a communication network, we can effectively infer the topological relationships between the radio stations. We hope that more people will be interested in research on LE behavior recognition of non-collaborative radio stations.

The remainder of this paper is organized as follows: Section 2 introduces the recognition algorithm model in detail and Section 3 introduces the experimental results and analysis. Finally, Section 4 shows our conclusion.

## 2. Methods

### 2.1. ACGAN

ACGAN, as a variant of Conditional GAN (CGAN), is widely used to generate fake “real” data. There are two modules in all different GAN models, which include a generator module and discriminator module. In the game against each other between the generator and discriminator during the training, the generator and discriminator can reach the ideal state. Then the generator can generate fake samples which are very similar to the real samples. The differences between ACGAN and CGAN are that ACGAN not only uses information of data’s labels for training, but also provides the category judgment of different samples. The structures of GAN and ACGAN are shown in Figure 1.

Compared with GAN, ACGAN generates more similar samples and it can also generate many samples with labels at a time. Therefore, ACGAN is very suitable for data augmentation and thus we are able to easily acquire more LE behavior signals with different labels.

During the training of ACGAN, the loss function of discriminator is expressed as:(1)$$L_{D}=L_{S}+L_{C}$$
(2)$$L_{S}=E[\log\;P(S=real|x_{real})]+E[\log\;P(S=fake|x_{fake})]$$
(3)$$L_{C}=E[\log\;P(C=c|x_{real})]+E[\log\;P(C=c|x_{fake})]$$

The loss function of the generator is expressed as:(4)$$L_{D}{}^{\prime }=L_{C}{}^{\prime }-L_{S}{}^{\prime }$$
(5)$$L_{C}{}^{\prime }=E[\log\;P(C=c|x_{fake})]L_{C}=E[\log\;P(C=c|x_{fake})]$$
(6)$$L_{S}{}^{\prime }=E[\log\;P(S=fake|x_{fake})]$$

During the training, the discriminator and generator alternately update parameters in the network model. The goals of network optimization are to maximize the $L_{D}$ of discriminator and $L_{D}{}^{\prime }$ of generator. In other words, the discriminator tries to distinguish the real samples from the fake samples, and the generator tries to make the generated samples be judged by the discriminator as the real samples. Finally, the loss function of the network model tends to be stable and the process of training is over.

### 2.2. Unidimensional ACGAN

Because it is often necessary to transform the electromagnetic signals into their deep features, the features are then treated as images. The LE behavior signals of a short-wave radio station cannot be directly put into a traditional ACGAN. The traditional method of signal recognition is to transform signals into their characteristic domain, and then these signals are processed as two-dimensional images.

However, the LE behavior signals of a short-wave communication station have few differences and their modulation is almost the same, except for the difference of the valid 26 bits. Thus, the characteristic transformations of LE behavior signals are incapable of getting better performance in recognizing LE behavior signals. Therefore, a new unidimensional ACGAN is proposed in this work. The unidimensional ACGAN was trained directly by the unidimensional LE behavior signals to achieve the purpose of getting more LE behavior signals with labels.

The structure of generator in unidimensional ACGAN is shown in Table 1.

As shown in Table 1, BN (0.8) represents the Batch Normalization (BN) layer, and the momentum is equal to 0.8. UpSampling1D means the data is upsampled by 2 times. Conv1D (Kernel Size (KS) = 3,1 (s)) denotes unidimensional convolution operation, the Kernel Size (KS) of which is 3 and convolutional stride is 1. Activation (‘*‘) denotes that the activation function is *.

The structure of the discriminator in unidimensional ACGAN is shown in Table 2.

As shown in Table 2, Conv1D (KS = 3,2 (s)) denotes unidimensional convolution operation, the Kernel Size (KS) of which is 3 and convolutional stride is 2. Activation (“LeakyReLU(0.2)”) means that the activation function is LeakyReLU(γ = 0.2). 7(class) represents the class of sample output by the discriminator.

Up to this point, the proposed unidimensional ACGAN has been shown in detail in Table 1 and Table 2. For LE behavior signals of a short-wave radio station, unidimensional ACGAN could be used to generate some data samples with different labels, and the original samples and generated samples could be combined to obtain a new training set. Then unidimensional DenseNet could be trained to effectively recognize different LE behaviors of a radio station. The method in this work is able to improve the accuracy of signals recognition.

### 2.3. LE Behavior Recognition Algorithm Based on Unidimensional ACGAN+DenseNet

When the number of LE behavior signals with labels of a short-wave radio station is relatively small, ta combination of unidimensional ACGAN and unidimensional DenseNet can improve the accuracy of LE behavior recognition. Firstly, the powerful generative adversarial ability of unidimensional ACGAN was used to generate fake “real” samples according to original training samples. Then real samples and generated fake “real” samples were mixed to form a new training set. The unidimensional DenseNet we proposed in [6] was able to effectively automatically extract the deep features of LE behavior signals, and finally Softmax classifier was used to realize the recognition of a radio station’s different LE behaviors. The whole process of the algorithm’s recognition did not need the help of the communication protocol, which provided a new idea for electronic reconnaissance. The framework of the recognition algorithm is shown in Figure 2.

The steps of the recognition algorithm are as follows:

**Step 1:** Firstly, a small number of samples with labels expressed as **X** are randomly selected from the real samples. Another sample expressed as **Y**, the dimensions of which are (500, 1), are generated by random noise, and these samples are randomly labeled as 0, 1, 2, 3, 4, 5, and 6, corresponding to seven kinds of LE behaviors.

**Step 2:** The **X** and the **Y** are put into unidimensional ACGAN as training data set. Unidimensional AGCAN begins to be trained. These parameters in generators and discriminators are updated alternately.

**Step 3:** According to the epoch and batch size, which we have set, repeat **Steps 1 and 2**.

**Step 4:** Randomly generate some noise sequences with specific labels, and then input these sequences into unidimensional ACGAN that have been already trained. Some fake samples with specific labels, expressed as **Z**, are also generated. Then a new training set is formed by mixing the real samples **X** with the generated samples **Z**.

**Step 5:** The batch size and epoch are set properly, and the new training set is used to train unidimensional DenseNet.

**Step 6:** A short-wave radio station’s LE behaviors are recognized by the trained unidimensional DenseNet.

## 3. Experimental Results and Analysis

Experimental environment: Intel (R) Core (TM) i9-9900K CPU, NVIDIA RTX TITAN×1, TensorFlow 1.12.0, and Keras 2.2.5.

### 3.1. LE Behavior Signals Dataset

According to the third-generation short-wave communication protocol standard (MIL-STD-188-141B), seven kinds of LE behavior signals only differ in the valid bits (26 bits) in their data frame. The LE behavior signals used in experiments are generated as shown in Figure 3.

As shown in Figure 3, TLC/AGC represent the Transmit Level Control process and Automatic Gain Control process. The short-wave communication protocol standard (MIL-STD-188-141B) stipulates that the carrier frequency is 1800 Hz, and the raised cosine filter is used to form waveform. The seven kinds of LE behavior signals, the dimensions of which were (5888, 1), were obtained, and the size of each behavior signal’s dimension was (5888, 1). In fact, 14,000 LE behavior signals were simulated and the number of every kind of LE behavior signal was 2000. Finally, the LE behavior signal dataset was ready for experiments.

### 3.2. Unidimensional ACGAN Generates LE Behavior Signals

There were a total of 14,000 signals that we simulated, which belonged to seven kinds of different LE behavior signals. Seven hundred signals were randomly selected from 14,000 signals, and they were treated as training sets of unidimensional ACGAN. We also selected 700 real samples as a validation set, and another 6300 samples formed the test set. An Adam optimizer was used in the experiments, the initial learning rate was 0.0002, momentum was 0.5, and batch size was 32. These LE behavior signals with SNR = 0 dB were put into unidimensional ACGAN.

The epoch was set as 20,000 when the unidimensional ACGAN was already trained adequately. The value of the discriminator’s loss and the value of the generator’s loss changed with training time going, as shown in Figure 4.

As shown in Figure 4, in the initial stage of network training, the loss of the discriminator is in the descending state, while the loss function of the generator is in the rising state. This indicates that the fake samples generated by the generator could not deceive the discriminator, while the discriminator could learn the deep features of the real samples and correctly distinguish the real samples from the fake samples. As a network model going to be trained adequately, the loss of the discriminator started to rise and the loss of the generator started to decline, which indicates that the fake samples generated by the generator were approaching the distribution of the real samples, and it was difficult for the discriminator to distinguish the real samples from the generated samples correctly. Then, the loss of the discriminator was in the descending state again and the loss of the generator was in the rising state again, indicating that the generator and the discriminator of the network model were fighting with each other in the training process. The generator gradually generated more real fake samples. Finally, the loss of the discriminator and the generator gradually became stable. The loss of the discriminator decreased slightly and the loss of the generator increased slightly, indicating that the samples generated by the generator could better approach the distribution of the real samples.

The Call behavior signal was one of the seven kinds of LE behavior signals of a short-wave radio station. As the number of epochs increases, the changes of the Call behavior signal generated by the unidimensional ACGAN’s generator are roughly shown in Figure 5.

According to Figure 5, with the process of training going on, after the value of epoch is greater than 2000, the Call behavior signal generated by unidimensional ACGAN is close to the real Call behavior signal. When the value of epoch was equal to 20,000, unidimensional ACGAN could generate fake “real” LE behavior signals. When the value of epoch is equal to 20,000, seven kinds of LE behavior signals generated by the network are shown in Figure 6.

As shown in Figure 5 and Figure 6, these samples generated by unidimensional ACGAN are similar to those real signals. However, whether there were differences between each kind of behavior signal requires more study on characteristic distribution of each kind of sample signal generated by our network. The unidimensional Convolutional Auto-Encoder (CAE) can effectively show the deep characteristic differences between different samples [26,27,28]. Therefore, the output of the auto-encoder was set as a two-dimensional vector. Differences between each kind of behavior signals are shown through the visualization of the two-dimensional vectors. The structures of unidimensional encoder and unidimensional decoder we proposed are shown in Table 3.

In Table 3, Conv1D represents one-dimensional convolution operation, S = 1 represents the stride of convolutional operation is 1, and KS represents that the size of the convolution kernel is 3. MaxPooling1D(2) represents that the stride of max-pooling is 2. And UpSampling1D represents that data is upsampled by two times.

The LE behavior signals whose SNR = 0 dB make up our dataset in our experiment. Firstly, 700 real signals selected at random were used to train the encoder, and then the fake signals generated by unidimensional ACGAN were put into the trained encoder. When the epoch of training unidimensional ACGAN was different, we received fake signals corresponding to different epoch with different labels (0, 1, 2, 3, 4, 5, 6) by unidimensional ACGAN. Then we put those fake samples into the CAE. The output of CAE is shown in Figure 7.

As shown in Figure 7, with the increase of the value of training epoch, the differences in characteristic distribution of the seven kinds of LE behavior signals become more and more obvious. According to Figure 7d, the distribution of yellow slightly overlaps the distribution of other colors because in the process of simulating signals labeled by 6 (6 means yellow), the 11 valid bits representing the called radio station’s address are generated randomly. This results in the similarities between Scan Call behavior signals and other LE behavior signals, and different LE behavior signals with SNR = 0 dB were only a little different. When the epoch was greater than 5000, the loss *g_loss* of the generator was still going up and down within a certain range, which made the characteristic distribution of the generated signals have a certain contingency. Comparing Figure 7c,d, with the training going on, the characteristic distribution of samples belonging to same class becomes more clustered. The experimental results show that unidimensional ACGAN could effectively generate different kinds of LE behavior signals, and unidimensional ACGAN could also learn the deep features of different kinds of behavior signals.

### 3.3. The LE Behavior Recognition Performance Based on Unidimensional ACGAN + DenseNet

In order to explore the performance of the unidimensional ACGAN+DenseNet algorithm for LE behavior recognition, we still randomly selected 700 samples from the simulated 14,000 LE behavior signals whose SNR was 0 dB. These 700 samples as the training set were put into unidimensional ACGAN. Then we selected 700 samples as the validation set and the 6300 samples were regarded as the test set. The original 700 samples were combined with fake samples generated by unidimensional ACGAN and then a new training set was acquired. The unidimensional DenseNet was be trained by the new training set. The value of the epoch for training the unidimensional DenseNet was set as 10, and the batch was set as 8. According to all the 50 Monte Carlo experiments, as the number of generated samples, which are used to train unidimensional ACGAN, changed, the average accuracy of unidimensional DenseNet changed, as shown in Figure 8.

As shown in Figure 8, when the number of the generated sample is 0, which means we used all real samples to train unidimensional DenseNet, the accuracy of recognition algorithm is 74.32%. As the unidimensional ACGAN generated more samples, the performance of the recognition algorithm was better. When the number of generated samples was 150, 200, 250, 500 or 1000, the accuracy of the algorithm model was improved by 1.24, 2.57, 4.32, 4.66, and 5.93%, respectively, compared to when the number of generated samples was 0. The experimental results show that the recognition performance of the network model could be improved if the samples generated by unidimensional ACGAN were combined with the real training data set.

In order to adequately verify the performance of our algorithm based on unidimensional ACGAN+DenseNet, especially under the condition that there were only few samples with labels, we set SNR = 0 dB and we used a different number of real samples to train unidimensional ACGAN. Seven hundred fake samples were generated by unidimensional ACGAN and then a new training set was acquired by combining 700 fake samples and the original training samples. There were 6300 samples still used as test set. According to all the 50 Monte Carlo experiments, as the number of original real samples which were used to train unidimensional ACGAN changed, the average recognition accuracy of unidimensional ACGAN+DenseNet algorithm changed, as shown in Figure 9.

As shown in Figure 9, if the number of generated samples is fixed at 700, and the recognition accuracy is 79.63% when the number of original training samples is 700. When the number of original training samples increased to 1000, 1300, 1600, or 1900, the recognition accuracy of the network increased to 83.32, 84.73, 86.12, and 90.06%. The experimental results show that the more original training samples we used, the better performance of unidimensional DenseNet would be, which is consistent with the real scene.

In order to further verify that more original training samples, which are used to train unidimensional ACGAN, were able to improve the performance of the recognition algorithm, the number of generated samples was no longer fixed at 700, and the number of generated samples was the same as the number of original training samples. Finally, according to all the 50 Monte Carlo experiments, as the number of original real samples, which are used to train unidimensional ACGAN, changed, the average recognition accuracy of the algorithm changed, as shown in Figure 10.

As shown in Figure 10, when the number of original training samples is 1000, 1300, 1600, or 1900, unidimensional ACGAN generates 1000, 1300, 1600, and 1900 LE behavior signals, respectively. Then, the new training set was used to train the network model and the final recognition accuracy increased as the number of training samples increased. When the number of original training samples was 1900, compared to when the number of original training samples was 700, the recognition accuracy of network model was improved by 10.49%.

Comparing Figure 9 and Figure 10, we can see that when the number of original training samples are the same, the number of generated samples will affect the final recognition accuracy of the network model. When the number of original samples was small, the difference in the number of generated samples will have a great influence on the recognition accuracy of the network model. For example, when the number of original training samples is 1000, the recognition accuracy in Figure 9 is 83.32%, while the recognition accuracy in Figure 10 is 84.34%. However, when the number of original training samples was large, the number of generated samples had a small impact on the final recognition accuracy of the network model, such as the number of original training samples being 1900. In summary, when training samples were sufficient, these samples could fully represent the essential features of these samples so that there was no need to utilize too many generated samples to train the network. However, it was necessary to generate more samples to train the network when the number of training samples was small.

### 3.4. Comparison Experiment

In order to illustrate the advantages of the unidimensional ACGAN+DenseNet algorithm, the proposed method was compared with Multilayer Perceptron (MLP), Lenet-5 [29,30], and unidimensional DenseNet [6]. They were all used to recognize different LE behaviors. 

The internal structure of MLP is as follows: Input layer--512--256--7--Output layer; “512”, “256”, and “7” represent the number of neurons in each hidden layer. The internal structure of Lenet-5 is as follows: Input layer--Conv1D(6,5,1(s))--MaxPooling(2)--Conv1D(16,5,1(s))--Maxpooling(2)--flatten--256--7--Output layer; “6” and “16” represent the number of convolution kernel; “5” represents the size of convolution kernel; “1(s)” represents that the stride of convolutional operation is 1; “MaxPooling(2)” represents that the stride of max-pooling is 2; “flatten” represents the flatten layer. The structure of unidimensional DenseNet is the same as the DenseNet used in this work.

From 14,000 LE behavior signals with SNR = 0 dB, 400, 700, 1000, and 1300 samples were randomly selected as the training sets, respectively, and 6300 LE behavior signals were selected as the test set. The number of original real training samples used to train unidimensional ACGAN and the recognition accuracy of each algorithm are shown in Figure 11.

As shown in Figure 11, when the number of original training samples is 400, 700, 1000, or 1300, the recognition accuracy of the algorithm in this work is higher than that of unidimensional DenseNet, LeNet-5, and MLP. In particular, when there were insufficient samples with labels, the recognition accuracy of LeNet-5 and MLP was less than 50%. When the number of training samples was 400, 700, 1000, or 1300, the recognition accuracy of unidimensional DenseNet was 1.92, 6.16, 4.63, and 3.06% lower than that of unidimensional ACGAN+DenseNet, respectively. It can also be known from Figure 11 that when the number of original training samples is large, such as 1300, the recognition accuracy of unidimensional ACGAN+DenseNet begins to be close to that of unidimensional DenseNet.

## 4. Conclusions

In the field of electronic countermeasures, not only is there no help of short-wave communication protocol standard, but it also is difficult to obtain a large number of LE behavior signals with labels. Therefore, it is a difficult problem to recognize the LE behaviors of a short-wave radio station. Firstly, according to the third-generation short-wave communication protocol standard, we simulated a radio station’s LE behavior signals with labels on behalf of labeled 3G ALE signals. Then a novel unidimensional ACGAN was proposed. A small number of samples with labels were used to train unidimensional ACGAN and some fake samples were generated. Then a new training set was formed by combining original real samples with generated samples, and the new training set was used to train the unidimensional DenseNet. Finally, the unidimensional DenseNet that had been trained was used to recognize different LE behaviors of a short-wave radio station. The experimental results show that the more training samples and the more fake samples generated were used to train unidimensional ACGAN, the better performance the network model had. Meanwhile, the experimental results showed that the performance of unidimensional ACGAN+DenseNet was much better than that of LeNet-5 and MLP. Moreover, when the number of original training samples was 400, 700, 1000, or 1300, the recognition accuracy of the algorithm in this work was 1.92, 6.16, 4.63, and 3.06% higher than that of unidimensional DenseNet. 

In summary, the proposed algorithm can recognize a non-collaborative radio station’s LE behaviors without the help of the communication protocol standard. In particular, when the number of labeled training samples was very small, such as 700, 1000, 1300, 1600, and 1900, the recognition accuracy of the algorithm could reach 79.63, 84.34, 85.23, 86.44, and 90.12%, respectively. In addition, a new unidimensional CAE was presented to explain the reliability of samples generated by ACGAN. Further, in terms of the application of the proposed algorithm in real time, we used a few LE behavior signals collected by sensors to train networks and the trained networks were used to recognize the new detected LE signals. When applying the algorithm, we did not need to consider the training time of networks, but the testing time of networks which was very short in reality.

In future work, the algorithm model of unidimensional ACGAN+DenseNet can continue to be optimized. In particular, the unidimensional ACGAN needs to be improved so that it can generate more real unidimensional electromagnetic signals, and other better data-augmentation techniques should be utilized to generate LE behavior signals; hence, there would be more contribution in terms of signals’ data augmentation. In addition, the unidimensional ACGAN was adopted in this work to directly generate time-domain signals. ACGAN was used to process features after signals’ characteristic transformation and then directly generate “real” features of signals to train the recognition network, which may have made the algorithm more productive under the condition that there were only a few labeled samples. Moreover, some study on real samples should be presented, as experiments are currently all based on simulated samples. As soon as conditions permit, the LE behavior signals of a short-wave radio station should be collected in a real environment by non-collaborative sensors and then the real samples should bde utilized to verify the performance of the proposed algorithm. Although the structure of the proposed algorithm should be further improved, it already has the capability to recognize radio signals, which demonstrates that we can recognize LE behaviors of radio station even without the communication protocol. Finally, research should be done on whether our idea could be applied to analyze communication behaviors and LE behaviors of other types of radio stations.

## Figures and Tables

**Figure 1 sensors-20-04270-f001:**
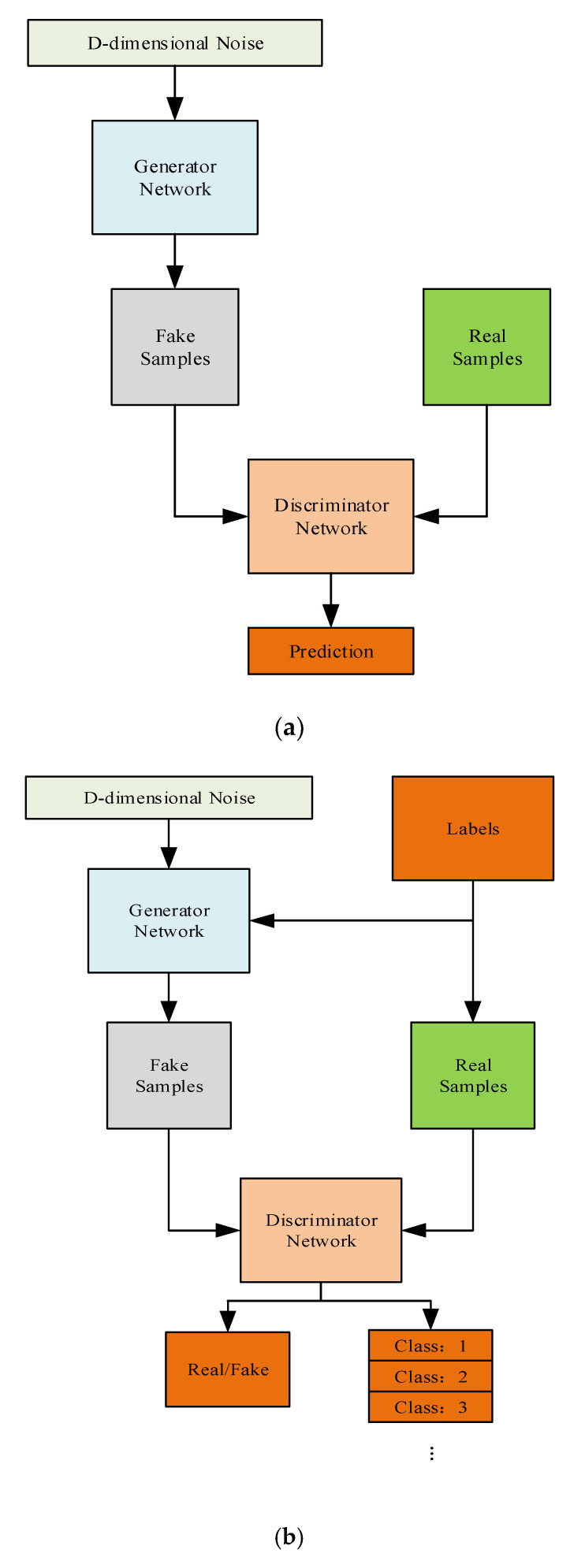
Structures of GAN and ACGAN. (**a**) Structure of original GAN; (**b**) structure of original ACGAN.

**Figure 2 sensors-20-04270-f002:**
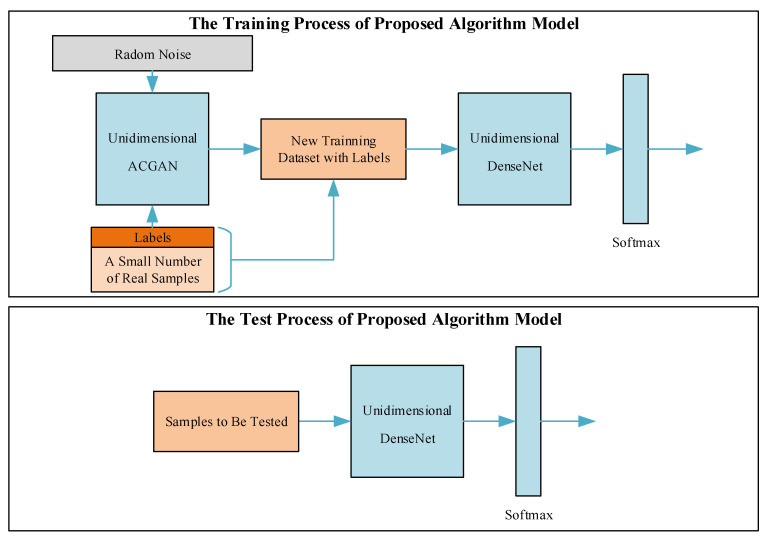
The framework of a short-wave radio station’s LE behavior recognition algorithm.

**Figure 3 sensors-20-04270-f003:**
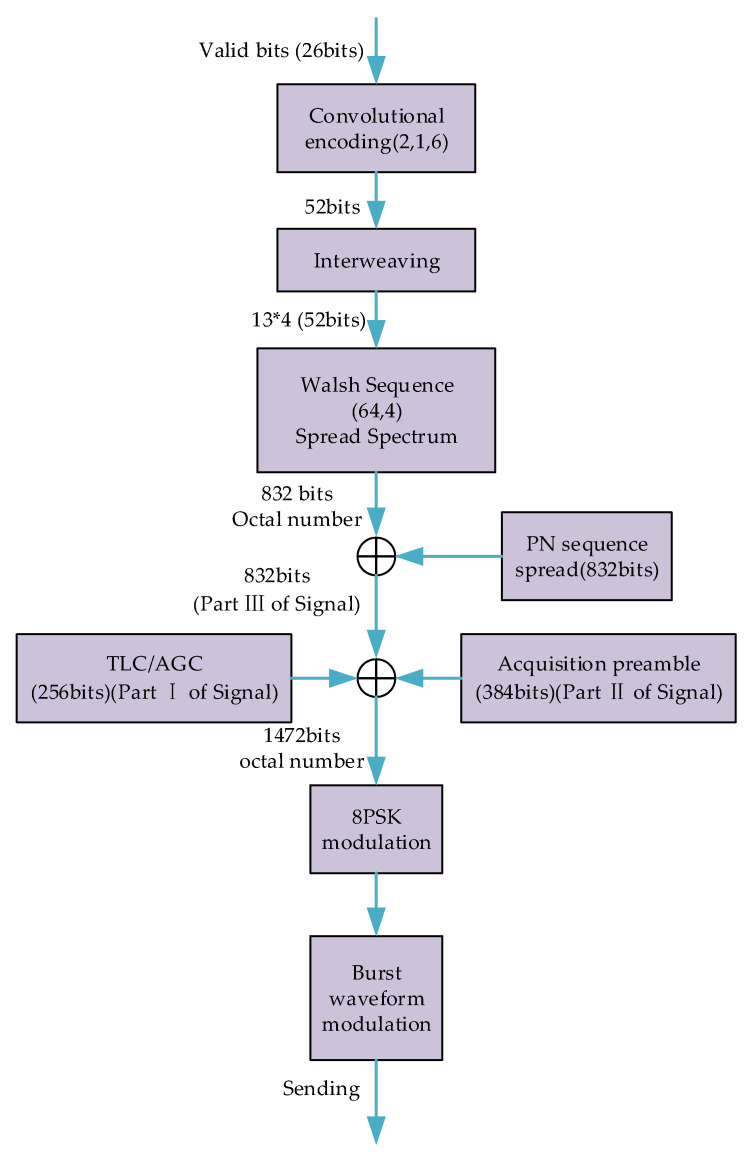
How the LE behavior signals used in experiments are generated.

**Figure 4 sensors-20-04270-f004:**
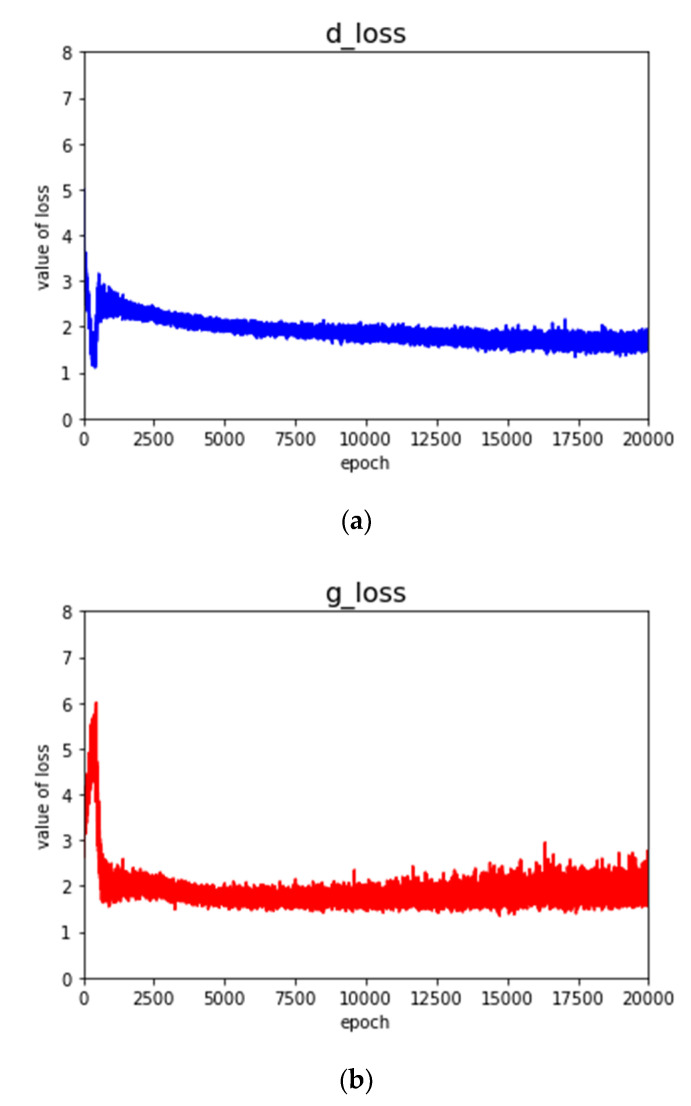
The curve of the loss function’ change. (**a**) Discriminator; (**b**) generator.

**Figure 5 sensors-20-04270-f005:**
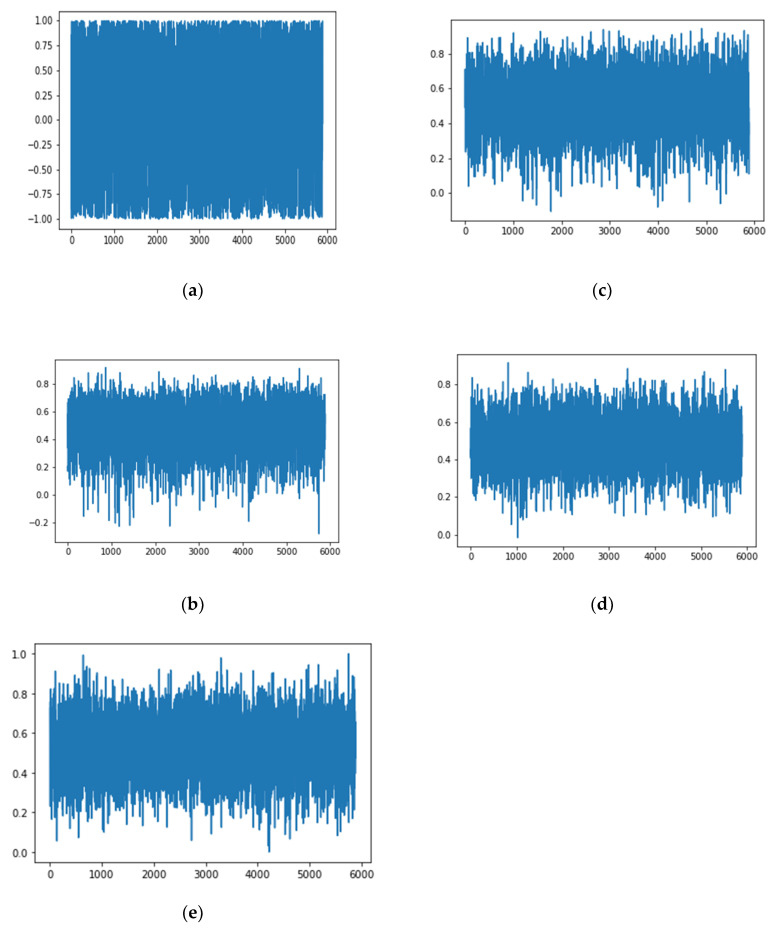
With different training epoch, the changes of the Call behavior signal generated by generator. (**a**) When epoch is 200; (**b**) when epoch is 1000; (**c**) when epoch is 2000; (**d**) when epoch is 20,000; (**e**) a real Call behavior signal.

**Figure 6 sensors-20-04270-f006:**
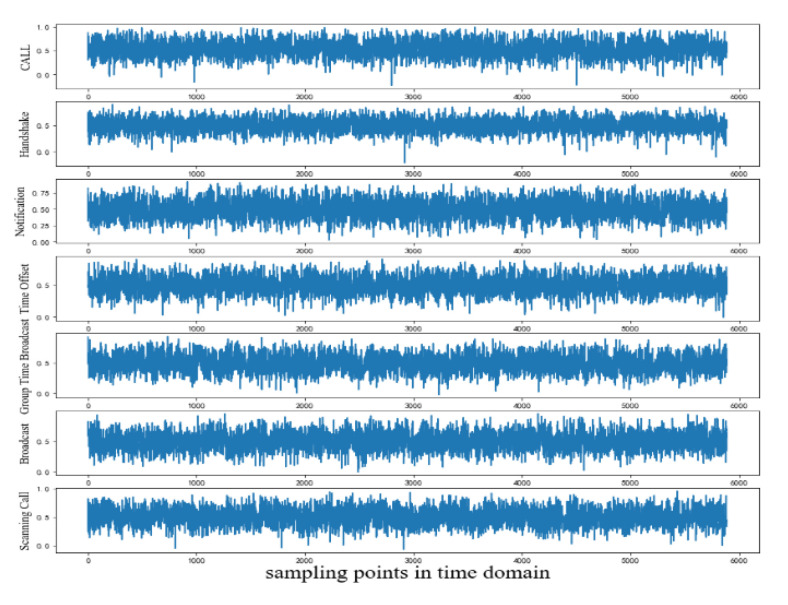
Seven kinds of LE behavior signals generated by unidimensional ACGAN.

**Figure 7 sensors-20-04270-f007:**
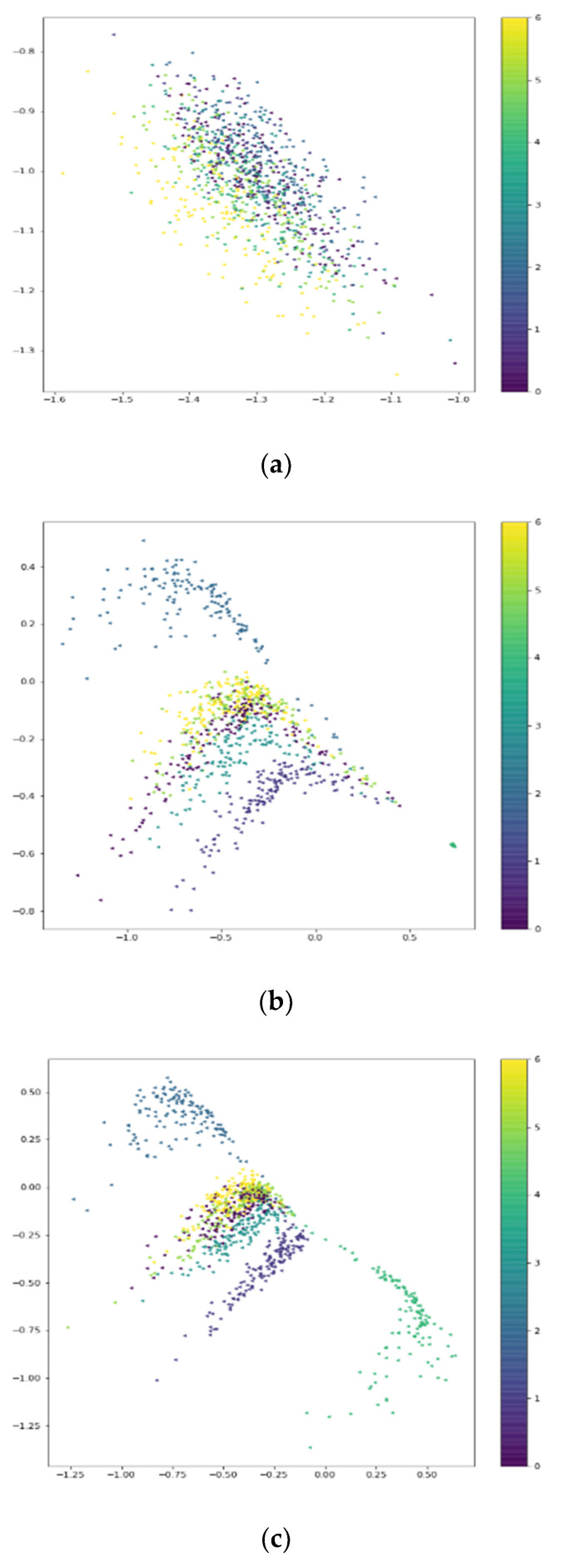
The outputs of unidimensional Convolutional Auto-Encoder (CAE) corresponding to generated signals with different training epochs. (**a**) When epoch = 500; (**b**) when epoch = 2000; (**c**) when epoch = 5000; (**d**) when epoch = 20,000.

**Figure 8 sensors-20-04270-f008:**
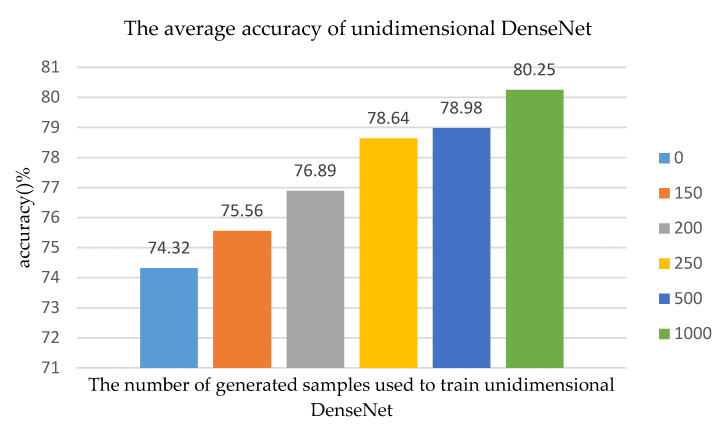
As the number of generated samples used to train unidimensional ACGAN changes, the average accuracy of unidimensional DenseNet changes.

**Figure 9 sensors-20-04270-f009:**
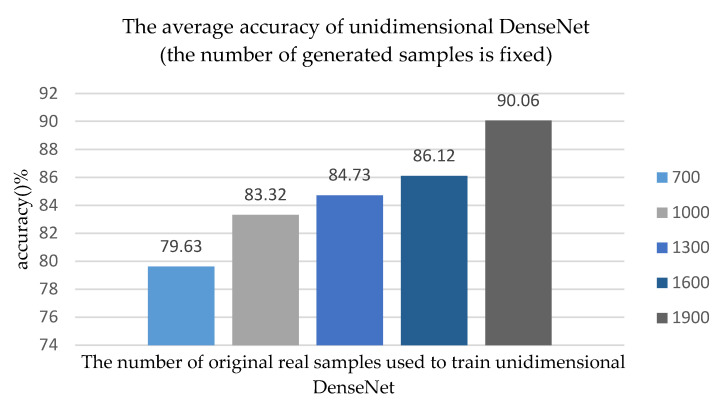
As the number of original real samples used to train unidimensional ACGAN changes, the average accuracy of unidimensional DenseNet changes, and the number of generated samples is fixed.

**Figure 10 sensors-20-04270-f010:**
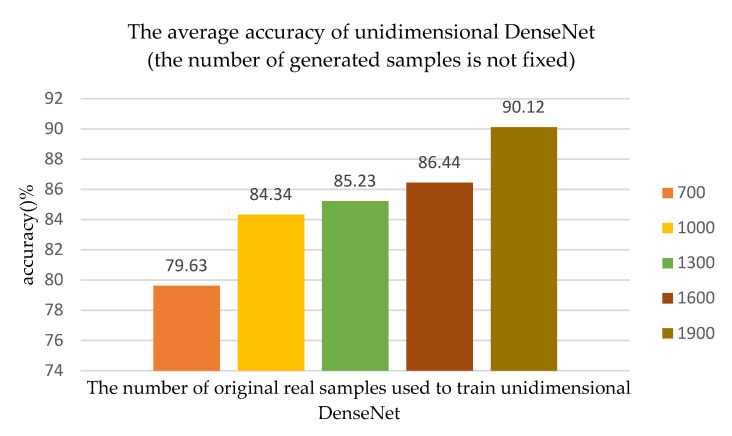
As the number of original real samples used to train unidimensional ACGAN changes, the average accuracy of unidimensional DenseNet changes, and the number of generated samples is the same as the number of original training samples.

**Figure 11 sensors-20-04270-f011:**
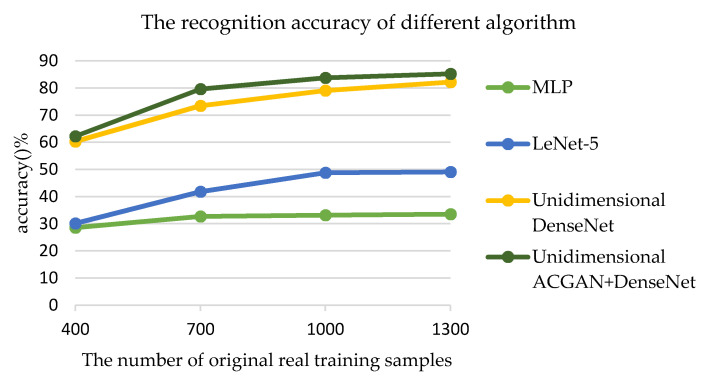
The number of original real training samples used to train unidimensional ACGAN changes and the recognition accuracy of the four algorithms.

**Table 1 sensors-20-04270-t001:** The structure of generator in unidimensional ACGAN.

Layer	Input Size	Output Size
Input	500(Noise)+1(Label)	500
Fully Connected	500	1472 * 30
Reshape	1472 * 30	(1472, 30)
BN(0.8)	(1472, 30)	(1472, 30)
UpSampling1D	(1472, 30)	(2944, 30)
Conv1D(KS = 3,1(s))	(2944, 30)	(2944, 30)
Activation(“ReLU”)+BN(0.8)	(2944, 30)	(2944, 30)
UpSampling1D	(2944, 30)	(5888, 30)
Conv1D(KS = 3,1(s))	(5888, 30)	(5888, 20)
Activation(“ReLU”)+BN(0.8)	(5888, 20)	(5888, 20)
Conv1D(KS = 3,1(s))	(5888, 20)	(5888, 1)
Output(Activation(“tanh”))	(5888, 1)	(5888, 1)

**Table 2 sensors-20-04270-t002:** The structure of discriminator in unidimensional ACGAN.

Layer	Input Size	Output Size
Input	(5888, 1)	(5888, 1)
Conv1D(KS = 3,2(s))	(5888, 1)	(2944, 20)
Actication(“LeakyReLU(0.2)”)	(2944, 20)	(2944, 20)
Dropout(0.25)	(2944, 20)	(2944, 20)
Conv1D(KS = 3,2(s))	(2944, 20)	(1472, 20)
Actication(“LeakyReLU(0.2)”)	(1472, 20)	(1472, 20)
Dropout(0.25)	(1472, 20)	(1472, 20)
Conv1D(KS = 3,2(s))	(1472, 20)	(736, 30)
Actication(“LeakyReLU(0.2)”)	(736, 30)	(736, 30)
Dropout(0.25)	(736, 30)	(736, 30)
Conv1D(KS = 3,2(s))	(736, 30)	(368, 30)
Actication(“LeakyReLU(0.2)”)	(368, 30)	(368, 30)
Dropout(0.25)	(368, 30)	(368, 30)
Flatten	(368, 30)	368 * 30
Output	368 * 30	1(real/fake)
		7(class)

**Table 3 sensors-20-04270-t003:** The structures of encoder and decoder.

**Encoder**		
**Layers**	**Input Size**	**Output Size**
Input Layer	(5888, 1)	(5888, 1)
Conv1D, S = 1, KS = 3	(5888, 1)	(5888, 20)
MaxPooling1D(2)	(5888, 20)	(2944, 20)
Conv1D, S = 1, KS = 3	(2944, 20)	(2944, 20)
MaxPooling1D(2)	(2944, 20)	(1472, 20)
Flatten	(1472, 20)	29, 440
Fully Connected	29440	256
(Out Layer)Fully Connected	256	2
**Decoder**		
**Layers**	**Input Size**	**Output Size**
Input Layer	2	2
Fully Connected	2	256
Fully Connected	256	29440
Reshape	29440	(1472, 20)
UpSampling1D	(1472, 20)	(2944, 20)
Conv1D, S = 1, KS = 3	(2944,20)	(2944,20)
UpSampling1D	(2944,20)	(5888,20)
Conv1D, S = 1, KS = 3	(5888,20)	(5888,20)
(Out Layer)Conv1D, S = 1, KS = 3	(5888, 20)	(5888, 1)

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
