# Peer review of "Unidimensional ACGAN Applied to Link Establishment Behaviors Recognition of a Short-Wave Radio Station"

_sensors, 2020, doi:10.3390/s20154270_

Round 1

Reviewer 1 Report

This paper presents a study of ACGAN + DenseNet for Link-establishment behaviors recognition. The proposed approach is validated on a simulated dataset. I have the following concerns for the paper:

  1. The novelty is limited. ACGAN is known to be able to generate more samples, and indeed the authors use it for data argumentation. The authors also aim to show the performance improvement of such a data-argumentation techniques. In terms of data argumentation, there is no significant contribution.
  2. The research has very little readership. It is not a research area of large research interests.
  3. The experimental evaluations are all based on simulated or generated samples. The authors should present some study on real samples.
  4. The comparison is not fair. The authors should try to use more samples in training. It is known that a large number of samples are needed to train a CNN, whereas the authors use as low as 700 samples for training, whereas use 7000 samples as validation set. The authors should first make good use of training samples, e.g. 10-fold cross validation. Then the authors consider to use data argumentation techniques for more samples.
  5. Based on the experimental results in Fig 9 and 10, it does not always increase the accuracy when more generated samples are used in the experiments, e.g. When 1900 simulated samples are used, using 700 generated samples works better than using 1900 generated samples.
  6. Fig 11 is a mess. The labels are all wrong.

Author Response

Response to Reviewer 1 Comments

Point 1: The novelty is limited. ACGAN is known to be able to generate more samples, and indeed the authors use it for data argumentation. The authors also aim to show the performance improvement of such a data-argumentation techniques. In terms of data argumentation, there is no significant contribution

 Response 1: Thank you very much for your comments, which guide us to make better achievements. Firstly, as far as we know, the ACGAN is rarely used in the field of electromagnetic signals, especially unidimensional signals. The original ACGAN is improved into a network that can directly process and generate unidimensional signals. Secondly, without the communication protocol standard, the proposed unidimensional ACGAN+DenseNet algorithm can still recognize LE behaviors, which means a lot in the military field. Lastly, maybe there is no significant contribution in terms of data augmentation. But we apply data-augmentation techniques to the field of radio station’ behaviors recognition. And we improve the performance of undimensional DenseNet so that LE behaviors can be recognized correctly by ACGAN+DenseNet even if there are only few samples. In addition, our focus is on whether we can recognize LE behaviors by utilizing very few labelled signals and the data-augmentation techniques indeed help in solving the problem as reviewer mentioned. In the future, we will continue to improve the data-augmentation techniques so that we can obtain more real unidimensional electromagnetic signals. Reviewer's comments give us new impetus.

According to reviewer’s Point 1, we have updated the manuscript at line 112-126 (“Our main contributions are …… non-collaborative radio station”) and line 450-453 (“Especially, the unidimensional …… signals data augmentation”).

Point 2: The research has very little readership. It is not a research area of large research interests.

Response 2: First of all, we think reviewer’s Point 2 is considerate. Researchers study on data augmentation may not pay too much attention to our research as reviewer mentioned. Researchers in the military field are very concerned about how to recognize LE behaviors of non-collaborative radio stations, which help the researchers infer network topology of these radio stations. For example, if we find that the behavior of a radio station is Call behavior, some other radio stations that communicate with the radio station appear after a period of time. Hence, all the above radio stations belong to the same communication network. Also, we can infer how many radio stations in the current communication network by analyzing the newly emerged electromagnetic signals. Moreover, other researchers specializing in signal classification are also interested in our work, especially when there are only few labelled signals. Our work can provide new ideas for electromagnetic signals classification under the condition of few samples. Therefore, we believe there are a lot of people interested in our research. However, due to confidentiality and difficulty, it seems that there are not so many research results. According to reviewer’s Point 2, we have modified the Introduction and Conclusions of the article and updated the manuscript at line 25-34 (“In the field of electronic …… emerged electromagnetic signals”) and line 453-456 (“In addition, the unidimensional …… few labelled samples”).

Point 3: The experimental evaluations are all based on simulated or generated samples. The authors should present some study on real samples.

Response 3: Reviewer’s Point 3 matters a lot. At the beginning of the study, we plan to collect real LE behaviors signals. However, due to the emergence of the COVID-19, this work is delayed. Once the epidemic is over, we will continue to collect real signals of non-collaborative radio station. As a matter of fact, we can only show part of the experiment results based on real LE behaviors signals due to confidentiality agreement. But we think this article can prove that it is possible to recognize LE behaviors of non-collaborative radio station without communication protocol, even though there are a few labelled signals. We promise that we will definitely carry out research based on real signals. According to reviewer’s Point 3, we have updated the manuscript at line 457-460 (“Moreover, some study …… the proposed algorithm”).

Point 4: The comparison is not fair. The authors should try to use more samples in training. It is known that a large number of samples are needed to train a CNN, whereas the authors use as low as 700 samples for training, whereas use 7000 samples as validation set. The authors should first make good use of training samples, e.g. 10-fold cross validation. Then the authors consider to use data argumentation techniques for more samples.

Response 4: Thanks for reviewer’s comment. As reviewer points, when there are lots of labelled samples, we really should use more samples in training, which we have done in the published paper entitled ‘Recognizing Automatic Link Establishment Behaviors of a Short-Wave Radio Station by an Improved Unidimensional DenseNet’. However, the application background of this work is that there are a few labelled samples, which is more common and difficult. Of course, it was unreasonable that we used 7000 samples as validation set. We should use fewer samples rather than 7000 samples. So we also select other 700 samples as validation set. We agree with reviewer’s opinion that 10-fold cross validation can make use of training samples, adequately so we adopt 10-fold cross validation in our experiments. Finally, according to Point 4, we conducted experiments again. The manuscript is updated in abstract, subsection 3.3 and subsection 3.4 at line 20 (“1.92%, 6.16%, 4.63% and 3.06%” ), line 334 (“700 samples”), line 344-349 (“ As shown in …… sample is 0”), line 364-367 (“As shown in ……86.12% and 90.06%”), line 383-393 (“When the number ……samples being 1900”), line 418-420 (‘When the number ……ACGAN+DenseNet, respectively’), line 437-438 (“Moreover, when the ……4.63% and 3.06%”).

Point 5: Based on the experimental results in Fig 9 and 10, it does not always increase the accuracy when more generated samples are used in the experiments, e.g. When 1900 simulated samples are used, using 700 generated samples works better than using 1900 generated samples.

Response 5: In the process of dealing with Point 4, we did experiments again. We find that the problem in Point 5 disappear. We think the reason for problem in Point 5 is that when the number of training samples is sufficient, using too many generated samples to train the network will cause the network performance to fluctuate. In fact, even in new experiments results, we can still find that as the number of training samples increases, the performance of the network only increases slightly though using more generated samples to train the network. It also means that when training samples are sufficient, these samples can fully represent the essential features of these samples. And if too many fake samples are used to train network, it may cause the network to extract some fake features of samples. We have updated the manuscript at line 393-396 (“In summary, when …… samples is small”).

Point 6: Fig 11 is a mess. The labels are all wrong.

Response 6: Thanks to the reviewer for helping us find the error. We are very sorry for this error, and now this error has been corrected. We have updated the manuscript at line 412-414 (“Figure 11 …… in figure”).

Reviewer 2 Report

This paper proposesa novel unidimensional ACGAN to get more signal samples and then use unidimensional DenseNetto realize recognizing link establishment behaviors. Firstly, a small number of real samples arerandomly selected from a large number of real link establishment behaviors signals as the trainingset of unidimensional ACGAN. Then, some fake samples are generated by unidimensional ACGAN.A new training set isformed by combining fake samples with real samples. A unidimensionalconvolutional autocoder is proposed to describe the reliability of the generated samples. Finally, this papercompletesthe recognition of different link establishment behaviors without the communicationprotocol standard, by using the new training set to train unidimensional DenseNet.

The topic is interesting and some novelty results were obtained. The detailed comments are as follows:

1.The language is generally well. However, there still exist some typos and grammar errors, which should be eliminated in the revision.

2. The main contributions of this paper are not clear, and please clarify the main contributions by comparing with the existing results in the revision.

3.Please clarify the motivation for considering the link establishment behaviors recognitionin the revision.

4.To increase the readability of this paper, the authors should simplify the abstract in the revision.

5. The explanation of short wave radio stationis not clear, and it needs to be clarified further.

6. For improving the implementation issue, please give a discussion if it is possible to generalize the current results to the short wave radio stationwith constraints, and for this issue, the following papers can be included in the revision to improve the literature review:command filter-based adaptive NN control for MIMO nonlinear systems with full state constraints and actuator hysteresis; observer-based fuzzy adaptive event-triggered control for pure-feedback nonlinear systems with prescribed performance.7.The main features of the proposed method are suggestedto be summarized in conclusions.

Author Response

Response to Reviewer 2 Comments

Point 1: The language is generally well. However, there still exist some typos and grammar errors, which should be eliminated in the revision.

 Response 1: Thanks for reviewer’s comments. We checked the manuscript again and eliminated several typos and grammar errors. Some typos and grammar errors in the manuscript have been corrected at line 40 (“are”), line 47 (“wants, with”), line 48 (“stations, wants”), line 52 (“stations which need execute these commands”) and so on, which are highlighted in yellow.

Point 2: The main contributions of this paper are not clear, and please clarify the main contributions by comparing with the existing results in the revision.

Response 2: The main contributions of this paper are listed at line112-126 (“Our main contributions are …… radio sation”). And we clarify some other contributions by comparing with the existing results in section 4 at line 437-443 (“Moreover, when the …… 90.12%, respectively.”).

Point 3: Please clarify the motivation for considering the link establishment behaviors recognition in the revision.

Response 3: We are sorry that the motivation is not clearly expressed. We further explained our motivation is in section 1 Introduction. The manuscript has been updated at line 28-34 (“Actually, researchers in …… electromagnetic signals”) and line 55-60 (“In addition, knowing …… low tactical status”).

Point 4: To increase the readability of this paper, the authors should simplify the abstract in the revision.

Response 4: Thanks for reviewer’s comment. The abstract of this paper has been simplified, which increases the readability. We have updated this manuscript at line 9-20 (“It is difficult …… uidimensional DenseNet”).

Point 5: The explanation of short wave radio station is not clear, and it needs to be clarified further.

Response 5: According to reviewer’s Point 5, we explained the short-wave radio in the article. And we have updated our manuscript at line 35-40 (“The short-wave radio station …… to intelligence reconnaissance”).

Point 6: For improving the implementation issue, please give a discussion if it is possible to generalize the current results to the short wave radio station with constraints, and for this issue, the following papers can be included in the revision to improve the literature review: command filter-based adaptive NN control for MIMO nonlinear systems with full state constraints and actuator hysteresis; observer-based fuzzy adaptive event-triggered control for pure-feedback nonlinear systems with prescribed performance.

Response 6: Thank you very much for reviewer’s advice. These two papers provide some ideas for the practical application of our algorithm, which is very important for us. We have updated the manuscript at line 75-77 (“In terms of …… in this work [7-8]”) and line 487-490 (“7.Qiu……2152-2162”).

Point 7: The main features of the proposed method are suggested to be summarized in conclusions.

Response 7: We appreciate reviewer’s help very much. The suggestion in Point 7 make this manuscript better. The manuscript has been updated at line 440-444 (“In summary, the …… generated by ACGAN”).

Reviewer 3 Report

From a mathematical point of view, the proposed algorithm is very good, but the applicability from the point of view of testing must still be justified.
The signal used for tests (laboratory simulation) is 3G ALE according to the MIL-STD-188-141B standard. This standard features frequency hopping and burst communication. Nothing is specified about the low probability of interception specific to these signals (SIGINT - Signal Intelligence), it is not mentioned which wave (BW0-BW5) of the 5 specified by the standard are simulated. The Conclusions mention the applicability of the proposed algorithm for real-time use, but do not mention whether for the 3G ALE signal, possibly other simpler HF radio signals in terms of structure.
The authors should mention why they chose the 3G ALE signal which has a very high complexity in terms of interception and did not test the algorithm on an HF signal with a lower radio complexity. The authors can also mention how they will apply the algorithm in real time (short description in Conclusions).

Author Response

Response to Reviewer 3 Comments

Point 1: From a mathematical point of view, the proposed algorithm is very good, but the applicability from the point of view of testing must still be justified

Response 1: We appreciate for reviewer’s Point 1 which matters a lot for our work. The focus of our work is to verify whether the LE behaviors of radio station could be recognized based on physical layer signals collected by non-collaborative sensors. Now the experimental results show that our ideas works. So as mentioned by reviewer, we will explore the applicability of the proposed algorithm in real environment later. We are convinced that the proposed algorithm can stand up to the test of real environment due that the signals used in this work are simulated in strict accordance with the protocol standard. We have updated the manuscript at line 426-428 (“Firstly, according …… 3G ALE signals”) and line 458-463 (“As soon as conditions …… communication protocol”).

Point 2: The signal used for tests (laboratory simulation) is 3G ALE according to the MIL-STD-188-141B standard. This standard features frequency hopping and burst communication. Nothing is specified about the low probability of interception specific to these signals (SIGINT - Signal Intelligence), it is not mentioned which wave (BW0-BW5) of the 5 specified by the standard are simulated. The Conclusions mention the applicability of the proposed algorithm for real-time use, but do not mention whether for the 3G ALE signal, possibly other simpler HF radio signals in terms of structure.

 Response 2: Thank reviewer very much for comments. Firstly, as reviewer said, it is a little difficult to intercept LE behaviors signals due to frequency hopping and burst communication. Therefore, in this paper, we only use a few samples to train the network, and then achieve the purpose of behaviors recognition. Also, our focus is on whether the LE behaviors of radio station can be recognized when the communication protocol is unknown. If the proposed algorithm works, it shows that we can continue to explore how to capture behaviors signals of the radio station. It is a very big project which includes intercepting LE behaviors signals and then recognizing radio station's behavior, so we want to implement our project step by step. Secondly, all the LE behaviors are transmitted using BW0 waveform according to the standard, so BW0 specified are simulated. Finally, thanks to reviewer’s reminder, some supplementary explanations are added in the conclusion. The proposed algorithm can be applied to not only real-time use but also 3G ALE signals. Actually, the algorithm will be evaluated on collected HF radio signals in the future. In our opinions, although the structure of proposed algorithm should be further improved, now it already has the capability to recognize radio signals, which can demonstrate that we can recognize LE behaviors of radio station even without the communication protocol. The manuscript has been updated at line 426-428 (“Firstly, according …… 3G ALE signals”) and line 458-463 (“As soon as conditions …… communication protocol”).

Point 3: The authors should mention why they chose the 3G ALE signal which has a very high complexity in terms of interception and did not test the algorithm on an HF signal with a lower radio complexity. The authors can also mention how they will apply the algorithm in real time (short description in Conclusions).

Response 3: The 3G ALE signal represents the beginning of radio station's communication. If we can recognize different ALE signals, we will know which behaviour the radio station want to conduct, which contains a lot of intelligence information. Also, the 3G ALE signal is widely used in military area. So we chose the 3G ALE signal even though it is complicated and difficult to be intercepted. In terms of the difficulty of interception, we have supposed that only a few labelled signals can be detected by sensors so that the ACGAN is utilized to improve the performance of recognition algorithm in this work. In addition, if the proposed algorithm can work based on 3G ALE signal, there is no doubt that it would have a good performance based on an HF signal with a lower radio complexity. In terms of the application of the algorithm in real time, we had used a few LE behaviors signals collected by sensors to train our networks and the trained networks would be used to recognize the new detected LE signals. When applying the algorithm, we do not need to consider the training time of the network but the testing time of the network which is very short in reality. On the basis of Point 3, we have updated the manuscript at line 25-34 (“In the field of …… electromagnetic signals”), line 42-45 (“In fact, the …… Scanning Call behaviour”) and line 444-448 (“Besides, in terms of application …… short in reality”). Thanks again to the reviewer for these suggestions.

Round 2

Reviewer 2 Report

This reviewer has no further comments.